# Big Data Handling Approach for Unauthorized Cloud Computing Access

Abdul Razaque [1,*], Nazerke Shaldanbayeva [1], Bandar Alotaibi [2,3,*], Munif Alotaibi [4,*], Akhmetov Murat [5] and Aziz Alotaibi [6]

1 Department of Cybersecurity, International Information Technology University, Almaty 050000, Kazakhstan; snazerke96@gmail.com
2 Sensor Networks and Cellular Systems Research Center, University of Tabuk, Tabuk 71491, Saudi Arabia
3 Department of Information Technology, University of Tabuk, Tabuk 71491, Saudi Arabia
4 Department of Computer Science, Shaqra University, Shaqra 11961, Saudi Arabia
5 Department of Information Security, L.N. Gumilyov Eurasian National University, Astana 010008, Kazakhstan; muratahmetov_1998@mail.ru
6 Department of Computer Science, College of Computers and Information Technology, Taif University, Taif 21944, Saudi Arabia; azotaibi@tu.edu.sa
* Correspondence: a.razaque@iitu.edu.kz (A.R.); b-alotaibi@ut.edu.sa (B.A.); munif@su.edu.sa (M.A.)

**Abstract:** Nowadays, cloud computing is one of the important and rapidly growing services; its capabilities and applications have been extended to various areas of life. Cloud computing systems face many security issues, such as scalability, integrity, confidentiality, unauthorized access, etc. An illegitimate intruder may gain access to a sensitive cloud computing system and use the data for inappropriate purposes, which may lead to losses in business or system damage. This paper proposes a hybrid unauthorized data handling (HUDH) scheme for big data in cloud computing. The HUDH scheme aims to restrict illegitimate users from accessing the cloud and to provide data security provisions. The proposed HUDH consists of three steps: data encryption, data access, and intrusion detection. The HUDH scheme involves three algorithms: advanced encryption standards (AES) for encryption, attribute-based access control (ABAC) for data access control, and hybrid intrusion detection (HID) for unauthorized access detection. The proposed scheme is implemented using the Python and Java languages. The testing results demonstrated that the HUDH scheme can delegate computation overhead to powerful cloud servers. User confidentiality, access privilege, and user secret key accountability can be attained with more than 97% accuracy.

**Keywords:** data security; data handling; access control; unauthorized access; cloud computing

## 1. Introduction

Cloud computing is a service that is currently popular [1]. Cloud computing companies provide almost all possible methods to process data: storing, changing, sharing with others, and eventually deleting [2,3]. However, the most attractive feature that distinguishes cloud computing from other traditional storage systems is its convenience; data can be obtained anywhere and anytime, instantly, with a connection to the internet [4]. People are looking for convenient, fast, and inexpensive systems to facilitate their tasks, which has become a factor in creation of the cloud systems, which respond to these needs but also face serious problems related to the security of data provided by users [5].

Depending on the services needed by organizations and individuals, cloud computing can be characterized by three existing models: software as a service (SaaS), platform as a service (PaaS), and infrastructure as a service (IaaS) [6–8]. SaaS includes software applications that are provided to customers for use in the cloud without any installation on their desktops, laptops, and so on. Today, various SaaS platforms around the world can be used [9]. The well-known examples of Saas platforms are Amazon's EC2 [10], Amazon's S3 [11], IBM's Blue Cloud [12], Google App Engine [13], Yahoo Pig, Google Apps [14],

Dropbox [15], and Salesforce's Customer Relation Management (CRM) system [16]. PaaS provides a platform for developing software applications provided by cloud computing, e.g., Amazon Web Services [10] and Window Azure [17]. IaaS comprises infrastructure, such as servers, operating systems, networks, and so on, that is provided to users through virtualization. Virtualization is the principal enabling core of cloud computing; it uses software to split one computer device into multiple independent computing devices, where each can be used to perform computing tasks. This helps to efficiently allocate and use the usually idle computing resources, reduces cost, and reliably increases infrastructure use. Among IaaS systems are DigitalOcean, Linode, Rackspace, Microsoft Azure, GCE, and so forth [18–20]. These systems significantly increase work efficiency in organizations at a relatively low price.

These organizations provide services in a pay-as-you-use manner at a relatively low price. For companies, it is easier to use cloud systems, because users can save the investments that would have been used for building their infrastructure [21].

In addition to these benefits, cloud computing has other advantages, such as increased efficiency, portability, scalability, and flexibility [22].

Cloud computing provides computing infrastructure resources over a network for organizations to use [23]. Despite their many benefits, cloud computing faces several challenges related to the security of access control and privilege management [24]. Potential threats can be addressed using machine learning algorithms [25].

Scalability, integrity, and data access are examples of the security and privacy challenges that face every cloud computing user. Data access is a significant aspect of the service of the cloud, without which the platform would not be able to operate or be so popular among users. Therefore, unauthorized access in a cloud system is one of the important problems that must be solved or prevented so that every user is able to trust the provider with their sensitive data [26].

Thus, the security and privacy of cloud data are important, as people and organizations are concerned that their data might fall into the hands of third parties who can use them for their own purposes [27]. For example, cloud data need to be secured within a trusted domain, e.g., data owners, industry and large organization data, and federal agents. Additionally, cloud data need to be saved in a fully trusted cloud database. Unauthorized data access is one of the existing security issues experienced by cloud computing systems [28] that must be solved and continuously reviewed.

Several mechanisms have been proposed to secure data access in cloud computing. These works have considered various problems in cloud computing and tried to achieve fine-grained, scalable, and secure data access control. For example, Yu et al. [18] combined three cryptographic techniques: KP-ABE, PRE, and lazy re-encryption. Each of them solves specific issues, creating an entire fine data access system. However, the system cannot properly handle multiple levels of attribute authorities. To solve this problem, Wan et al. [19] introduced a new approach that extended Yu et al.'s [18] solution. To achieve flexible and scalable data access control in cloud computing, Yu et al. [18] implemented a hierarchical attribute-set-based encryption (HASBE) algorithm. Wang et al. [20] combined two existing algorithms, hierarchical identity-based encryption (HIBE) and cipher-text policy attribute-based encryption (CP-ABE). After this, Wang et al. [20] investigated performance trade-offs and then used proxy and lazy re-encryption algorithms on the given output. Other researchers [29–33] proposed other solutions for secure access control, which are described in detail in Section 3. However, the proposed methods have their own disadvantages and shortcomings. For example, these methods are susceptible to several types of cyber threats. Some of them use public key encryption, which is slow compared to symmetric encryption and has many potential certification issues since it depends on a third party. Different from existing methods, we constructed a method called HUDH. The HUDH scheme provides the integration of three state-of-the-art algorithms for data encryption, data access, and intrusion detection. These algorithms successfully restrict unauthorized access in cloud computing systems. The proposed HUDH scheme

is practically applicable because the algorithms in the approach are already standardized, being used separately in business and market-oriented applications.

## 2. Problem Statement

Security is an essential issue for any computing environment [34], as the data, hardware, and software should be protected from unauthorized access. A cloud service provider in cloud computing offers computational services and virtualization over the Internet. It provides critical services such as restricting unauthorized access, maintaining data integrity, and ensuring data availability. To ensure cloud security, security challenges must be addressed to take full advantage of this computing paradigm [35]. To deeply analyze the origins of unauthorized access in cloud computing systems regarding causation, we followed a proper research methodology. The existing schemes were studied in accordance with research strategies to address the problem of unauthorized access in cloud computing. Thus, we found that an efficient model of access control including encryption, data access, and intrusion detection algorithms is required to secure data in the cloud, as well as to implement measures against intrusion [36].

Cloud computing involves an enormous number of devices, applications, and parties that make designing a secure data sharing framework a difficult task to accomplish. Moreover, data on the cloud are susceptible to various threats such as losses, accidental alteration by the cloud provider, and attacks [37]. Thus, developing a complete security method for cloud storage is necessary.

To achieve our aim, we reviewed the existing methods used in cloud computing systems and investigated the use of advanced security mechanisms such as advanced data encryption, secure data access, and accurate intrusion detection mechanism to build a secure cloud computing model. We then used a popular dataset to test and analyze the ability of the proposed model to resist many types of attacks.

### 2.1. Motivation

Building security mechanisms for cloud storage is an essential task. Legitimate participants who want to share their data on the cloud want secure control and access mechanisms, as well as fast and safe sharing of data on demand. The currently available methods have several shortcomings. Thus, we need a robust cloud computing system that can provide advanced data encryption, secure data access, and accurate intrusion detection mechanism.

### 2.2. Paper Contributions and Organization

In this paper, a mechanism for handling unauthorized data access in cloud computing is proposed consisting of several steps: data encryption, data access, and intrusion detection [38–40]. To implement these mechanisms, we used some existing efficient algorithms: advanced encryption standards (AES) for encryption, the attribute-based access control (ABAC) algorithm for data access control, and the hybrid intrusion detection (HID) algorithm for unauthorized access detection.

- The HUDU scheme integrates three state-of-the-art algorithms (ABAC, AES, and HIDS) to ensure data security, user authentication, and prevention of potential threats in cloud computing.
- The HIDS algorithm combines the features of two known algorithms (random forest and neural network) for important feature selection and training the data. A higher accuracy was achieved with the integration of these two algorithms.
- The proposed HUDU scheme was tested on the known UNSW-NB15 dataset using Class 4 and Class 6 to confirm its accuracy. The use of different classes provides a new direction for ensuring security, as higher accuracy was achieved when employing a higher class.

This paper is organized as follows: Section 3 reviews the existing schemes used to solve the unauthorized data access problem in cloud computing environments. Section 4

presents a detailed description of the proposed scheme (HUDH) involving three algorithms: AES for encryption, ABAC for data access control, and HID for unauthorized access detection. Section 5 shows the implementation of the method and presents the results in detail. Section 6 discusses the results. Section 7 concludes the paper.

## 3. Related Works

In this section, we review the existing schemes and models that have been proposed to solve the unauthorized data access problem in cloud computing. All these studies provided several methodologies for addressing this issue, which we describe in general. We then explain the advantages and disadvantages of their solutions.

Large-universe attribute-based encryption with public traceability for cloud storage was proposed by Zhang et al. [41]. This method mainly addresses both key abuse and key escrow concerns when deploying ABE in a cloud computing environment. Two different approaches are used: a key generation center (KGC) and an attribute authority (AA), which cooperate for the generation of the user's secret key. Both the KGC and AA do not know the full decryption key or have the capability to forge one. However, this model requires expensive computations in the cloud.

To achieve secure, fine-grained, and scalable data access on the cloud, Yu et al. [18] combined three leading cryptographic techniques, KP-ABE, PRE, and lazy re-encryption, and applied them to the data being stored in the cloud. A set of attributes were added to the data file to give each user access to the structure of that set of attributes in the form of a tree. To achieve this, the KP-ABE algorithm is used to escort the data encryption keys of data files. This facilitates the fine-grainedness of access control. However, this algorithm, if used alone, places a large computational overhead on the owner, because users are charged for the use of the given services. This overhead mostly comes from the user revocation operation, where the owner must re-encrypt all the data files accessible to the user who is leaving the system. The owner is also required to be online the entire time while the revocation is running. To solve this problem, a hybrid algorithm was introduced that included PRE and KP-ABE algorithms that allowed the owner to delegate the computational overhead to the cloud server. The server from its side cannot read the files because of the attributes in the files. This helps the owner to avoid the extra overhead and to control the data in the cloud. However, in this case, the server has computational overhead. To reduce this overhead, another algorithm, known as lazy re-encryption, was added, which helps the server to unite the tasks of some computational operations. The operation on the server is independent of the number of users because the computations are performed on the attributes in the data file or on the size of the access structure in the form of a tree. The number of users added or removed does not matter and the data access is not corrupted. Therefore, the algorithm can provide scalability. However, with these advantages, the problem of scalability arises for multiple levels of attribute authorities.

A novel digital forensic architecture was constructed using fast-growing software-defined networking (SDN) and blockchain technology for infrastructure as a service (IaaS) cloud [42]. The secure ring verification-based authentication (SRVA) scheme was proposed to protect the system from unauthorized users. To strengthen the cloud environment, secret keys are generated using the harmony search optimization (HSO) algorithm. For encryption, the sensitivity-aware deep elliptic curve cryptography (SA-DECC) algorithm is used.

A sensitive and energetic access control (SE-AC) mechanism was proposed for providing secure access control even in critical situations [43]. The mechanism ensures the confidentiality of data by authorizing individuals with limited permissions to edit or modify patient data. Data are encrypted before being sent to the cloud storage. The user can obtain the required access and their permissions are changed based on authentication and context attributes. In addition, the access operation to the IoT can be controlled and assessed for security analysis to prevent any unauthorized access.

A secret sharing group key management protocol (SSGK) to protect the communication process and shared data from unauthorized access was proposed [37]. Different from

the prior methods, a group key is used to encrypt the shared data, and a secret sharing scheme is used to distribute the group key in SSGK. Extensive security and performance analyses indicated that the protocol considerably minimizes the security and privacy risks of sharing data in cloud storage and saves about 12% of storage space.

A novel fog-centric secure cloud storage scheme to protect data against unauthorized access, modification, and destruction was presented by Han et al. [37]. To prevent illegitimate access, the proposed scheme employs a new technique known as XOR combination to conceal data. Moreover, block management outsources the outcomes of XOR combination to prevent malicious retrieval and ensure better recoverability in case of data loss. The proposed approach is based on a hash algorithm for the higher-sensitivity detection of malicious attacks. The robustness of the scheme was demonstrated through a security analysis and experimental results, validating the performance of the proposed scheme in terms of data processing and time.

Wan et al. [19] proposed the hierarchical attribute-based encryption (HASBE) algorithm, which is an extension of the attribute-based encryption algorithm with an added hierarchical structure delegation algorithm that is similar to the CP-ABE scheme. In addition, the security of HASBE was proven, because the CP-ABE algorithm, which is similar to the hierarchical structure, was secure under some models of vulnerabilities. Scalability was also achieved by extending the ASBE algorithm with a hierarchical structure. The algorithm was used to delegate the generation operation of the user's private attribute to the lower-level domains. Compared to the method proposed by Yu et al. [18], these methods provide flexibility by giving the organization of the user attributes a recursive structure, which allows the user to impose dynamic constraints on those attribute combinations to satisfy the policy. Therefore, the given algorithm can handle complex attributes and multiple operations for given attributes. Because HASBE is based on ASBE, the algorithm can provide fine-grained access to the users, as well as efficient user revocation by assigning attributes to each user's key and provide multiple value assignments of those attributes. Therefore, this algorithm is well-suited to solving the given issue.

Wang et al. [20] proposed an algorithm that helps companies to efficiently share data on cloud servers. This solution combines two existing algorithms: hierarchical identity-based encryption (HIBE) and cipher-text policy attribute-based encryption (CP-ABE) algorithms. The proposed method has a performance to expressivity trade-off and uses proxy and lazy re-encryption algorithms on the given output. This method has several advantages, such as high-quality performance, fine-grained access control to the system, scalability, and full delegation. The HABE scheme is known as collision-resistant, which has already been proven to be secure against random and adaptive attacks. This model includes a root master (RM), a third trusted party (TTP), domain masters (DMs), and the users, which are the personnel in an organization. The RM is used for the generation of the parameters and keys of the domain and the distribution of these results, whereas the DMs are used to delegate the keys to the next level and the users. In the HABE model, each DM and attribute with an ID are marked, while each user is assigned an ID as well as an attribute. After, Wang et al. [20] extracted the object's private key from the DM, where the public key is the combination of the public key and the ID of the DM. This algorithm shows how data can be efficiently delegated and shared in a cloud server.

Li et al. [29] proposed a novel framework for access control to PHRs within the cloud computing environment. To enable fine-grained and scalable access control for PHRs, they leveraged attribute-based encryption (ABE) techniques to encrypt each patient's PHR data. To reduce the complexity of key distribution, they divided the system into multiple security domains, where each domain manages only a subset of the users. As such, every user, i.e., the patient, can fully control the privacy of the given data, and the complexity of key management in the system is reduced. The proposed solution gives the system flexibility, efficiency in user revocation, and data access in emergencies. The advantages of the methodology include: markedly facilitating storage, allowing PHR service providers to shift their PHR applications and storage into the cloud, elastic resources,

and reduced operational cost. However, the method has a number of disadvantages: by storing PHRs in the cloud, the patients lose physical control of their health data, which makes it necessary for each patient to encrypt their PHR data before uploading to the cloud servers, and key distribution can be inconvenient when there are multiple owners, since it requires each owner to always be online. The authors also discussed methods for enabling efficient and on-demand revocation of users or attributes, and break-glass access under emergency scenarios. They leveraged attribute-based encryption (ABE) techniques to encrypt each patient's' PHR data. Additionally, they used public key cryptography (PKC)- and symmetric key cryptography (SKC)-based solutions.

Nurmi et al. [30] presented a framework, known as EUCALYPTUS, which is based on the infrastructure as a service (IaaS) model. This system provides the owner with virtual machine instances that are distributed in different physical resources. In the method with the precisely described EUCALYPTUS algorithm, the system's operational aspects and architectural structure directly affect the scheme, such as the algorithm's portability, modularity, and simplicity. Additionally, evidence showed states that EUCALYPTUS encourages users, who used systems such as Grid and HPC, to explore other functionalities of the system while working with it. Each component is implemented as a stand-alone web service, which helps make the system modular. This also provides the system with the following benefits: a well-defined API in the form of WSDL, which contains operations that the system can perform; input/output data structures; and the ability of users to use the existing features of the service for secure communications among components.

Khan et al. [31] reviewed various features of attribute-based access control mechanisms suitable for a cloud computing environment. This led to the design of an attribute-based access control mechanism for cloud computing. Access control decisions are important for any shared system. However, for cloud computing, factors such as scalability and flexibility are crucial. Various access control methods are used in cloud computing, which highlight the features of attribute-based access control that are important for its design. Many access control methods have been considered, analyzing their benefits and weaknesses. In identity-based access control models, including mandatory access control (MAC), discretionary access control (DAC), and role-based access control (RBAC), users (subjects) and resources (objects) are identified by unique names. The benefit of using these names is that it leads to these access control methods being effective in unchangeable distributed systems, where there are only a set of users with a known set of services. However, in large distributed open systems, users and resource providers are not placed in the same security domain. The users are characterized by their attributes and properties rather than by predefined identities. Therefore, identity-based access control models are not very effective in these systems. This is their main weakness. RBAC provides access to the user based on the job role. In RBAC, permissions can be added or deleted if the privileges for a role change, so the main advantage is dynamically changeable permissions. However, reaching an agreement regarding what privileges to associate with a role is difficult. In attribute-based access control (ABAC), access is granted based on attributes that the user can prove they have, such as date of birth or national number. Attribute-based access control is an extension of role-based access control, in general, with the following features: delegation of attribute authority, decentralization of attributes, and interference of attributes. ABAC is popular because it is flexible in supporting various kinds of domains and policies. Its disadvantages are due to two factors: reaching an agreement on a set of attributes is challenging, especially across multiple agencies or domains and organizations, and the existing methods of access are based on the authentication of the user from one site, as well as from the time the request is sent. Sometimes, this method is labeled as authentication-based access control. In all of these methods, the domains must be tightly coupled to combine identities or to identify the meaning of attributes and roles. Additionally, this approach makes it difficult to assign a set of administrator attributes to users. Zissis et al. [32] discussed various features of the attribute-based access control mechanisms that are suitable for cloud computing environments, leading to the design of a suitable method. ABAC provides policies for the sensitivity of the important data of the user. In addition, it

provides autonomy to the organization while working with it, and automates trust negotiation, which can be checked when required. Each object in the system has attributes that define the identity and characteristics of its respective object. These policies are supported by the authorization system of the server. Each system has its own policy description methods. To integrate policies into the system and to provide scalability to data access, each of these rules is encapsulated as an independent unit. This method was important for designing an attribute-based access control.

Hota et al. [33] introduced a technique based on the capability to solve data access control that only allows valid users to access to the data in the cloud. They also modified the Diffie–Hellman protocol for key exchange between the user and provider, to secretly share each other's symmetric keys. This provided a secure data access D–H key exchange model for cloud systems. The results of their analysis showed that the technique is efficient and secure on some existing security models; it also includes a set of security protocols that handles the data access control in the cloud infrastructure. Furthermore, the proposed solution includes a capability-based model as well as public-key encryption to protect sensitive data in the cloud. The users are provided a D–H key exchange model to securely access data from another domain. This solution does not use the user's public key, facilitating key management of the system. The ciphers of the public key, private key, and of the hash are located in an isolated and secure place in the cloud. The proposed approach also encourages the owners to store their sensitive data in the cloud without losing control of it. It also delegates the computational overhead to the server, making it more attractive to users.

Hahn et al. [44] introduced a secure cloud-based IoT data management model to efficiently manage cloud storage and bandwidth. The proposed model is also capable of tracking traitors who illegitimately reveal their secret keys to other parties. The proposed method consists of three phases: (1) each IoT user has a specific transformation key, and the key holder identity is authenticated in the cloud; (2) the ciphertext is partially decrypted utilizing the transformation key of the IoT user; and (3) eventually, the partially decrypted result is returned. Although this approach can securely and efficiently manage cloud storage, a cloud server must be installed to perform decryption and reduce the computation overhead associated with decryption.

In cloud computing, both blockchain and deep learning have shown success in many cases, particularly in improving the security of the cloud. For example, Alkadi et al. [45] used both blockchain and a deep learning algorithm named a bidirectional long short-term memory (BiLSTM) to improve data privacy and detect cyberattacks, Moreover, a deep learning algorithm was used to detect and identify any malicious malware or virus and the blockchain was used to provide more robust data privacy. However, BiLSTM, which has double LSTM cells, is known to be computationally heavy, costly, and difficult to train.

Kimmel et al. [46] used an RNN model to improve the security of cloud services, particularly against cloud malware. Cloud malware is software that is widely used to attack VMs hosted on cloud IaaS. The authors used two different types of RNN algorithms—bidirectional recurrent neural network (BRNN) and long short-term memory (LSTM)—to rapidly and effectively detect malware. The model was trained to detect the behavior of malware. It achieved a high accuracy of over 99%. However, both LSTM and BRNN require a large dataset to be trained.

Similarly, other deep learning techniques such as convolutional neural networks (CNN) [47,48] have been used for cloud security. Although deep learning techniques provide acceptable performance, a lighter weight technique with minimum complexity is needed for easy and suitable deployment in the cloud. Deep learning techniques have many layers, complex structures, and a huge number of neurons and parameters. Thus, they are computationally expensive and require strong computing power and large network capacity.

Namasudra [49] used three well-known techniques—attribute-based encryption (ABE), identity-based timed-release encryption, and distributed hash table (DHT)—to design a secure and efficient cloud computing access control method. The aim of this access control is

to provide cloud computing environments with resource- and knowledge-sharing. Initially, the method uses user attributes to encrypt the data. Then, the encrypted data are split into two parts: the extracted ciphertext and the encapsulated ciphertext. Consequentially, the ciphertext shares are generated by encrypting the decryption key and combining both the extracted ciphertext and the key's ciphertext. Finally, the author suggested distributing the ciphertext shares into the DHT network and storing the encapsulated ciphertext on cloud servers. The proposed approach proved its effectiveness compared to related methods through security and performance analysis. However, combining more encryption methods increases the computational complexity.

Namasudra et al. [50,51] proposed an access control method based on the popularity value to provide cloud computing users with efficient data access. The proposed approach relies on a public key to provide the requested data to users while decreasing the access time and search cost. Moreover, the security of data confidentiality for users is maintained. A significant aspect of this proposed approach is its capability to resist several attacks such as phishing, stolen verifier, masquerade, man-in-the middle, and internal attacks. However, public-key encryption is slower than symmetric encryption; thus, it would be preferable to use symmetric encryption because it is faster.

## 4. Proposed Hybrid Unauthorized Data Handling

In this study, we designed a mechanism that considers data access control from a different point of view. One algorithm protects the data if the intruder gains access by encrypting the data. The second algorithm handles the access issue and safely delegates rights to a user. Th final algorithm provides fast detection of intrusions to the system, which enables quick reactions.

When the data are loaded to the cloud, the system automatically applies the AES algorithm to encrypt and store the data in the database. Each data owner has a secret key and the consumer of that data has the public key. Therefore, if the data are in the possession of the third party, it is not possible to use the data, because there is no key to decipher them. For each user, data and actions have their own attributes, which will be used by the ABAC algorithm to find relationships among them and provide access if the policy allows it. The intrusion detection system works when the session of the user starts and continues monitoring until the session ends. If the system identifies anomalous behaviors of the user, it instantly sends a notification message to the administrator.

In Algorithm 1, Step 1 shows the initialization process of the used variables. The input and output processes are shown at the beginning of the algorithm. Steps 2–7 apply the AES cryptographic algorithm to obtain encrypted data by substituting bytes, shifting rows, and mixing columns. The time complexity of the encryption algorithm is $O(m + n)$, where $n$ is the text length and $m$ is the pattern length.

---

**Algorithm 1:** Encryption algorithm.

---

   **Input:** $\{D, I_k\}$ in
   **Output:** $\{D_e\}$ out
  1: **Initialization:** $\{D$: data, $D_b$: database, $D_e$: encrypted data, Ae: AES algorithm, N: number of iterations, $I_k$: initial key, $R_k$: round key, $B_r$: byte rows, $B_c$: byte columns, B: bytes$\}$
  2: **Do**
  3: **Add** $R_k$
  4: **Apply** *Ae* on *D* by shifting *B*
  5: **Shift** $B_r$
  6: **Mix** $B_c$
  7: **While** $N \cong 9$
  8: **Get** $D_e$
  9: **Store** $D_e \rightarrow D_b$

---

Step 8 shows the created encrypted data. Step 9 is the process of storing the encrypted data in the database.

In Algorithm 2, Step 1 shows the initialization process of the used variables. The input and output processes are shown at the beginning of the algorithm. Step 2 shows the process of defining policies in a JSON file. Step 3 makes a request to the JSON file to obtain access to data. Steps 4–10 show if a user has the right to the data, then access is granted, and data is obtained from the database and deciphered; otherwise, an error message is returned to the request, as defined in (1):

$$a(x) = \{03\}x^3 + \{01\}x^2 + \{01\}x + \{02\} \tag{1}$$

---

**Algorithm 2:** Data access algorithm.

---

   **Input:** $\{R_d\}$ in
   **Output:** $\{D_d, M_e\}$ out
  1: **Initialization:** $\{D_b$: database, $D_e$: encrypted data, $D_d$: deciphered data, $M_e$: message,
      $P_r$: policy, J: JSON file, $R_d$: request to data access, K: key for deciphering, $A_e$: AES algorithm$\}$
  2: **Set** $P_r \rightarrow J$
  3: $R_d \rightarrow J$
  4: **if** $R_d \cong A$ into $J$ **then**
  5:    **Get** $D_e$ from $D_b$
  6:    **Use** $K$ and $A_e$ on $D_e$
  7:    **return** $D_d$
  8: **else**
  9:    **return** $M_e$
10: **end if**

---

The encryption step involves the AES algorithm to obtain ciphertext. The most important part of this algorithm is the key expansion. The encryption consists of multiple rounds depending on the size of the key, and each round has a new key. The routine creates $4x(N_r + 1)$ words, where $N_r$ is the number of rounds. We can use a particular equation to calculate and find keys in each round easily, as defined in (2):

$$K[n] : s[i] = k[n-1] : s[i] \oplus k[n] : s[i] \tag{2}$$

where $k$ is the size of the key that consists of 16 bytes and $s$ represent every four bytes of that key.

For $s_0$, we have to use a particular equation that is different from the above equation as follows. These equations are used to find a key for each round rather than $s_0$, the key from our initial key, where $R_{con}[i]$ is the round constant for round $i$ of the key expansion:

$$\begin{aligned} K[n] = s_0 &= k[n-1] : s_0 \\ &\oplus SubByte(k[n-1] : s_3 \\ &\gg 8 \oplus R_{con}[i]) \end{aligned} \tag{3}$$

where $n$ is a constant that is equal to $0 \times 63$, $Arr$ is the initial bytes, and $x^r$ is the number of required rounds to find the inverse. $I$ is determined by the following nonlinear equation:

$$I = Arr + x^r + n \tag{4}$$

To calculate the number of rounds $n_r$, we use the following mathematical equation, as defined in (5):

$$n_r = \frac{S_k}{S_b} + 6 \tag{5}$$

where $S_k$ is the key size and $S_b$ is the block size.

To implement byte substitution, Equation (6) is used:

$$
\begin{aligned}
b_i &= b_i \oplus b_{(i+4)mod8} \oplus b_{(i+5)mod8} \\
&\oplus b_{(i+6)mod8} \oplus b_{(i+7)mod8} \oplus C_i
\end{aligned}
\tag{6}
$$

where $b_i$ is the $i$th bit of the given byte and $c_i$ is a bit of the given byte with a specific value. The time complexity of the data access algorithm in the best case is $O(logn)$; in the worst case, it is $O(n)$.

In Algorithm 3, Step 1 involves the initialization process of the used variables. The input and output processes are shown at the beginning of the algorithm. Step 2 involves the process of training our models using a dataset and previous log files.

---

**Algorithm 3:** Intrusion detection algorithm.

---

**Input:** $\{D_s, L_o\}$ in
**Output:** $\{M_e\}$ out
1: **Initialization:** $\{D_s$: *dataset, D: data, A: access, $M_e$: message, $L_o$: data in the log, U: user,*
   $P_a$: *patterns, S: active status, $M_{sd}$: supervised model, $M_{ud}$: unsupervised model, $N_l$: new log*$\}$
2: **Train** $M_{sd}$ and $M_{ud}$ using $D_s$ and $L_o$
3: **Do**
4: **Apply** $M_{sd}$ on $N_l$
5: **if** $N_l \cong P_a$ **then**
6:      **return** $M_e$
7: **else**
8:      **Apply** $M_{ud}$ on $N_l$
9: **end if**
10: **if** $N_l \neq L_o$ **then**
11:      **return** $M_e$
12: **end if**
13: **While** $U \cong S$
14: **End while**

---

Steps 3–13 check the current log of the user against known patterns and previous logs using trained models to check if the behavior of that user is malicious, then returns a warning message about intrusion to the administrator of the system; otherwise, it continues to monitor while the user is active. The space complexity of the intrusion detection algorithm could be slightly higher because it iteratively applies a depth-first search, which depends on the depth or length of the optimal path. Thus, the space complexity is $O(bm)$, where $m$ is the maximum depth and $b$ is the branching factor.

In order $S'_r$, which is the next step in the AES algorithm, we use Equation (7):

$$
\begin{aligned}
S'_{r,c} &= S_{r,(c+shift(r,Nb))modNb,} \\
&\quad for\ 0 < r < 4\ and\ 0 \le c \le Nb,
\end{aligned}
\tag{7}
$$

where the function $shift(r, Nb)$ is used to create the action of moving the byte to a lower position and is dependent on the row value, $s$ is the byte substitution table, $r$ is the row, $c$ is the column, and $Nb$ is the number of bytes.

For mixing the columns, a fixed polynomial is used, which is given by:

$$
a(x) = \{03\}x^3 + \{01\}x^2 + \{01\}x + \{02\},
\tag{8}
$$

Figure 1 shows our data encryption algorithms. As can be observed, the initial file is ciphered using the key. The key changes in each round of the AES process.

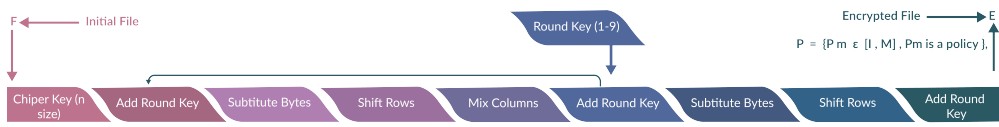

**Figure 1.** Data encryption with the AES algorithm.

Then, the byte substitution is implemented and the rows are shifted.

After this, the column will be mixed and the process starts again. We used a dataset to display the intrusion detection ability of the method. This dataset contains categorical and numerical features with various scales, which is why we needed to transform it into metric space, i.e., numerical values, and normalize the values. Below, we demonstrate this method.

Each categorical feature expressing the number $n$ of possible categorical values is transformed using a function $f$ that maps the $m$th value of the feature to the $m$th component of an $n$-dimensional vector:

$$f(x_i) = (0, \ldots, 1, \ldots, \ldots, n) \tag{9}$$

where $x_i$ is to the value of $m$. From the formula, the 1 in the braces is located at position $m$. Then, we apply these new features and the numerical features' scaling function by using their corresponding mean $\mu$ and standard deviation $\sigma$ values:

$$f(x_i) = \frac{x_i - \mu}{\sigma} \tag{10}$$

To create the neural network, we used the following formula for one neuron to calculate the hidden layer:

$$h(x) = \sigma\left(w_j + \sum_{i=1}^{n} w_{ij} x_i\right) \tag{11}$$

where $\sigma$ is a nonlinear activation function, $x_i$ is the initial $i$th node, and $w$ is the weight for that particular node. To calculate the output, we use the following equation:

$$o = \sum_{j=1}^{n} h_j w_j \tag{12}$$

where $h$ is a hidden layer and $w$ is the weight for that particular hidden layer.

The attributes of each entity define the identity and characteristics of its corresponding entity.

- $A_{req} = \{ReqAttr_i | i \in [1, I]\}$;
- $A_{serv} = \{ServAttr_j | j \in [1, j]\}$;
- $A_{res} = \{ResAttr_k | k \in [1, K]\}$;
- $A_{env} = \{EnvAttr_l | l \in [1, L]\}$, where $I$, $J$, $K$, and $L$ represent the maximum number of attributes per entity.

Security policies are defined in the system of the cloud. Each user may have their own defined policies. The policy $P$ supported by ABAC as a superset of these policies is defined in (13):

$$P = \{Pm \in [I, M], Pm \text{ is a policy}\}, \tag{13}$$

The access decision is made based on policy evaluation by the decision function $f$. $P_f$ is the evaluation function of policy $p_n$ and is defined as follows:

$$Pn_f(A_{req}, A_{serv}, A_{res}, A_{env}) = permit \text{ or } deny \tag{14}$$

Polices are evaluated by providing the attributes of the entities to the decision function $f$, defined as follows.

$$
\begin{aligned}
Decision_A BAC = f(&Requestor, \ Service, \\
&Resource, \ Environment) \\
P1_f(&Requestor)\&P2_f(Service)\& \\
P3_f(&Resource)\&P4_f(Environment),
\end{aligned}
\tag{15}
$$

The encrypted data are sent to the database. When the user requests the data, the ABAC algorithm is called. The algorithm uses the JSON file where all the policies and rules are defined.

The attributes of the user, action, and data are compared. When correspondence is found, access is granted to the user. The AES algorithm is applied again to decipher the encrypted data.

Figure 2 illustrates the data encryption and data access algorithms. The initial key is transformed to another form using key expansion. This key is then used to encrypt data using logical XOR.

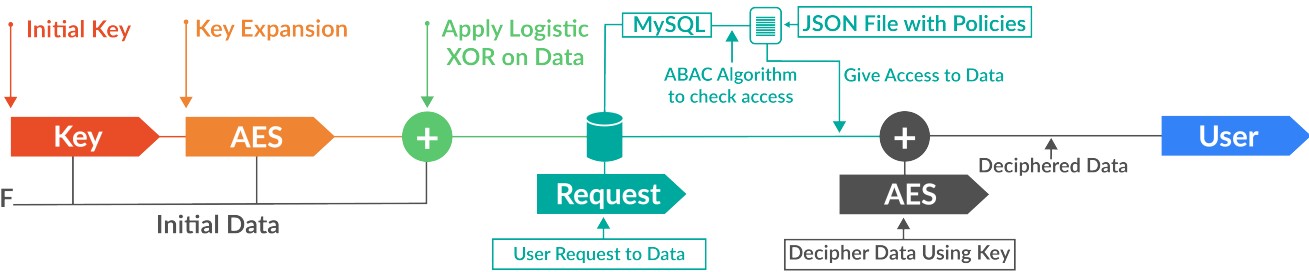

**Figure 2.** Data encryption and data access algorithms.

## 5. Implementation and Experimental Results

In this section, we describe the implementation of the mechanism. This mechanism was built on a MacOS Mojave operating system using PyCharm IDE in Python 3.4.6 language and IntelliJ IDE in Java. Additionally, we used JSON files to write policies and rules to implement the ABAC data access algorithm. All the data and user information were stored in a MySQL database. The existing UNSW-NB15 dataset [52] was used for IDS implementation. Normal and abnormal network traffic were generated in a real-world test bed. The abnormal traffic data were generated using hacking tools, which consist of nine types of attacks: denial-of-service, worms, shellcode, exploits, generic, fuzzers, backdoors, reconnaissance, and analysis. A ground truth table consisting of the nine types of simulated attacks was used to label the dataset. The dataset input was a set of instances. Each instance contained various data types such as float, integer, binary, and nominal data. The instances that belong to one of the nine types of simulated attacks were labeled as abnormal (i.e., one), and the normal instances were labeled as zero. The number of instances (i.e., the rows in two-dimensional space) in the dataset is 2,540,044, with 2,218,761 belonging to the normal class and the remaining 321,283 belonging to the abnormal class. The number of features (i.e., the columns in two-dimensional space) is 49, representing the network packet fields such as source IP address, destination IP address, source port number, destination port number, and so on. Table 1 shows the materials used for implementation.

**Table 1.** Materials Used.

| Item | Description |
|---|---|
| Platform | Python 3.4.6, Java |
| Operating system | MacOS Mojave |
| RAM | 8 GB |
| ROM | 128 GB |
| Database | MongoDB |
| Processor | 1.8 GHz Intel Core i5 |
| Data set | UNSW-NB15 |
| Policy list | JSON |
| Environment | PyCharm, IntelliJ IDE |

To justify the effectiveness of the proposed HUDH, several scenarios were generated that fell within two classes:

- Data access process by legitimate/malicious insider user;
- Data access process by outside user (malicious outsider).

*5.1. Legitimate/Malicious Insider*

Figure 3 shows the data access process. When the user tries to access data, the ABAC algorithm is activated. The algorithm involves the user, data, and action attributes from the database, as well as the policies from the JSON file to check the privileges and rights. IntelliJ IDEA is used to implement the ABAC algorithm while Apache Tomcat and Maven are used to run the process. The JSON file is used for data access.

When the user sends a request to the system, it is sent to the permission-evaluator component, where it is further delegated to the policy-enforcement component, which makes all the decisions based on the rules. The policy-definition component is used to load all the policy rules. After loading the policy enforcement, it compares the request against the rules and if the system returns true, access is granted. If a user attempts to access data for which the user does not have the rights, access is denied and the user is considered a malicious insider. Furthermore, the IDS algorithm constantly monitors the requests of the user; if it identifies anomalous activity, then the system administrator is notified.

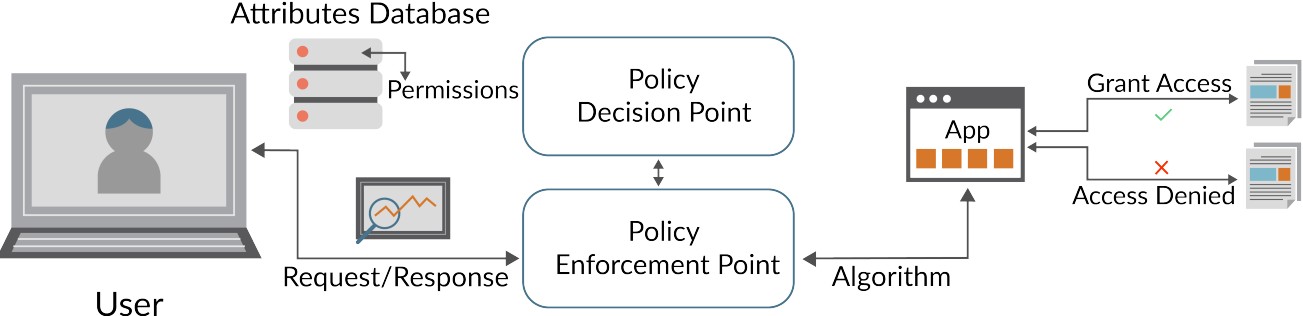

**Figure 3.** Data access mechanism.

*5.2. Malicious Outsider*

Figure 4 illustrates the case of an attack on the cloud system. The attacker uses the command-and-control method to obtain control to cloud providers, then it generates the attacks. In this case, 5000 attacks were generated. The cloud computing server has the support of the designed IDS algorithm, which continuously monitors for malicious activity and denies unusual requests.

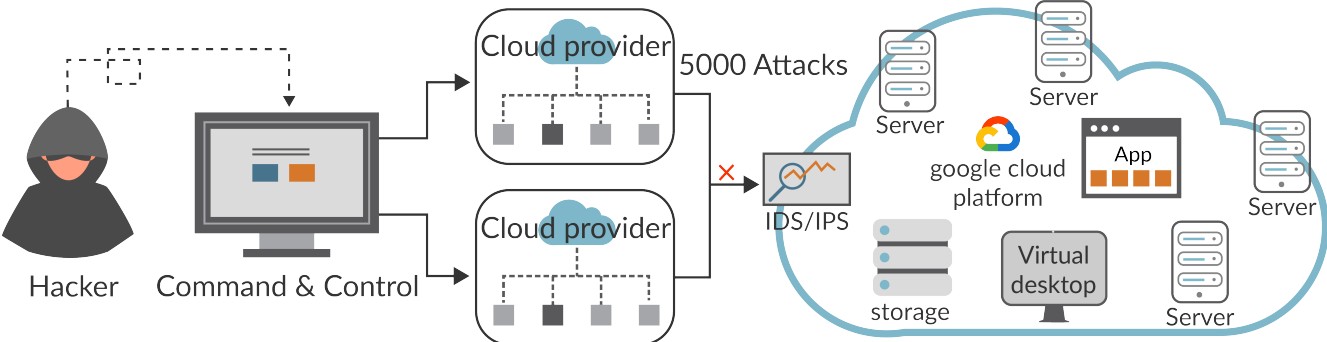

**Figure 4.** Attacks from outside the cloud.

*5.3. Results*

Based on the testing results, the following parameters were determined:

- Response time;
- Predicted/normal behavior;
- Accuracy.

### 5.3.1. Response Time

Figure 5a plots the result, which shows the time performance of each request in the administrator account. The scenario involves different kinds of requests of the system, as shown in Table 2. As can be seen, the performance for each request is different. We performed operations such as add, delete, view, list of users, and project tables. The operations for users' tables required more time. The reason for this is the complexity of the table; it has many attributes and relationships.

**Table 2.** Different types of tested requests.

| | |
|---|---|
| 1 | add project |
| 2 | list project |
| 3 | view project |
| 4 | delete project |
| 5 | delete user |
| 6 | add user |
| 7 | list user |

We observed different response times, which are due to the use of the administrator account because the administrator has more privileges. When the administrator makes requests, it involves the database and JSON file, whereas the project manager only receives a denial, which only involves the JSON file. The results demonstrated that the administrator account requires 9.67 ms to complete the process; the project manager account requires 25.48 ms to complete the process.

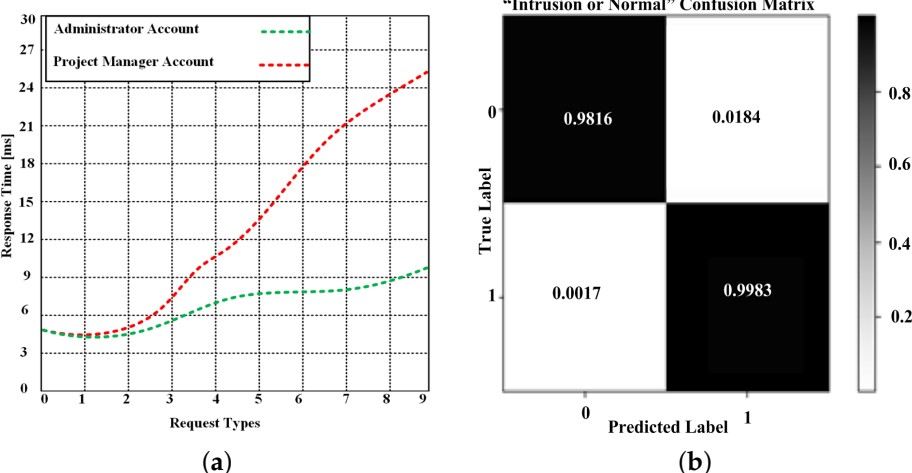

**Figure 5.** (**a**) Time required for administrator and project manager accounts. (**b**) Confusion matrix of the classification model.

### 5.3.2. Intrusion/Normal Behavior

Figure 5b shows the confusion matrix for the proposed HUDH. Based on the test results, normal and intrusion behaviors were determined. As shown by the confusion matrix, the proposed HUDH method has high prediction accuracy. It obtained 0.9816 correct answers and 0.0017 wrong answers for normal behavior and 0.9983 correct and 0.0184 wrong answers for intrusion behavior.

### 5.3.3. Accuracy

Figure 6a shows the feature importance of the random forest algorithm and Figure 6b demonstrates the feature importance for the neural network algorithm. The feature importance is chosen only for determining the required attributes to train the model. These chosen features mostly impact the definition of an intrusion to the system. We used the scikit-learn library to determine the feature importance in random forest as the RandomForestClassifier and RandomForestRegressor classes. The dataset is obtained from this link: "https://scikit-learn.org/stable/modules/generated/sklearn.datasets.make_classification.html (access on 4 December 2021)" make_classification() function was used to generate the test dataset. The dataset contained 1000 examples with input features. Of the input features, 50% were informative and 50% were redundant. The feature_importances_property was retrieved to obtain the importance scores for each input feature. The system uses random forest to classify data as normal or malicious data.

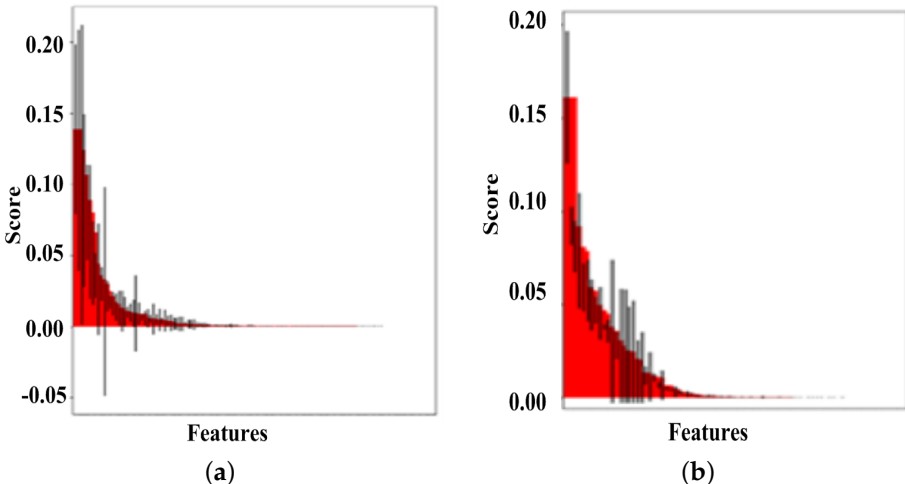

**Figure 6.** Feature importance for (**a**) random forest and (**b**) neural network.

The resulting information was then further used to train the neural network to classify the attack data based on the different attack categories. As such, we increased the accuracy of the scheme. We confirmed that the HIDS scheme works correctly by testing it. All the access decisions are based on the rules defined within the default policy (JSON file). All users are defined in the Memory_User_Details_Service file.

Figure 7a displays the receiver operating characteristic (ROC) curve of the classification model. It shows the ability of the classification model to diagnose the given request, based on the confusion matrix results. In summary, it shows the accuracy of the trained model. The ROC is equal to 0.9962, which is a good result, indicating the model can be used to predict whether a user is authorized by integrating different environments.

Figure 7b demonstrates the history of the neural network model. The loss decreased and the model accuracy increased as the number of epochs increased. The accuracy increased 0.36% and loss reduced approximately 0.47%. Based on this result, we determined that proposed model produces high-accuracy performance.

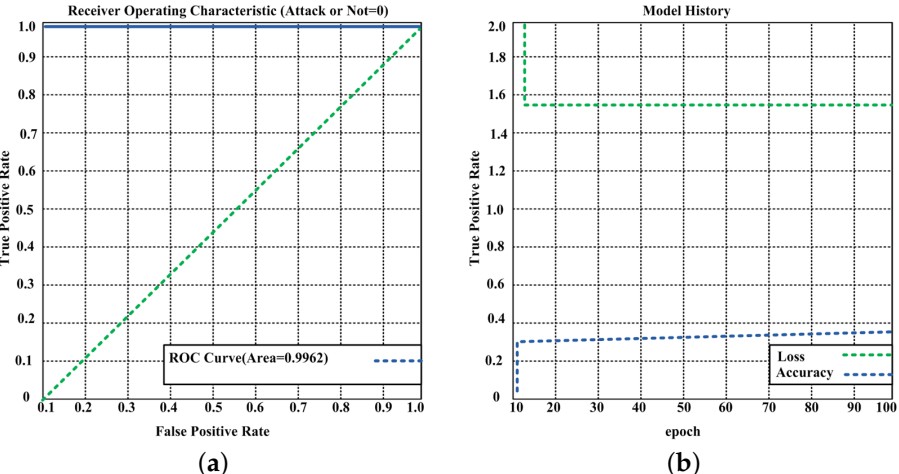

**Figure 7.** (**a**) ROC curve of the classification model. (**b**) Neural network model history.

Figure 8a illustrates the ROC curve of our neural network model for Class 6, which shows a 0.9975 accuracy. Classes 4, 2, and 0 are also considered important in the feature importance step. We only show Classes 6 and 4 for comparison purposes.

Figure 8b illustrates the ROC curve of our neural network model for Class 4, which shows a 0.9710 accuracy for this class, which had a high importance in the previous step. Based on the results, we can see that Figure 8a,b use a different class. Thus, they have different accuracy ratios. If a class is higher, then higher accuracy can be obtained.

The average accuracy of the HUDH scheme was determined with a maximum 5000 generated threats and compared to state-of-the-art algorithms: SA-DECC [42], SE-AC [43], BRNN-L, and IDTRE [46,49]. Based on the results (Figure 9), we found that the proposed HUDH method produced an average accuracy of 99.48%; the average accuracies of BRNN-L, IDTRE, SE-AC, and SA-DECC were 98.51%, 97.82%, 98.14%, and 97.79%, respectively, showing that the proposed scheme has higher average accuracy.

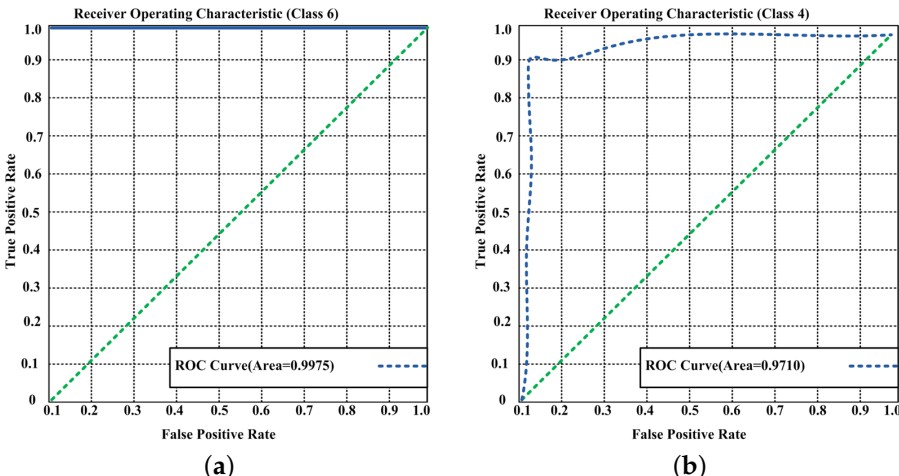

**Figure 8.** ROC curve for (**a**) Class 6 and (**b**) Class 4.

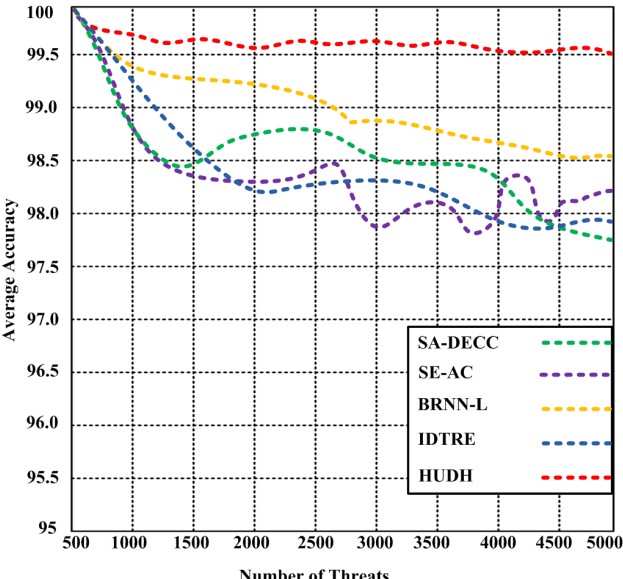

**Figure 9.** Average accuracy of the proposed HUDH and other state-of-the-art methods: IDTRE, BRNN-L, SE-AC, and SA-DECC, with a maximum of 5000 threats.

## 6. Discussion

The proposed HUDH scheme applies access control to reduce the possibility of unauthorized data access, enhances the accuracy of the intrusion detection system, and provides more protection by encrypting the data in the cloud. Table 3 lists the security capabilities of our scheme compared to those of the others. Table 3 shows that the proposed HUDH provides higher accuracy by integrating three algorithms. In comparison, the other methods possess either one or two features that lower their accuracy. The HUDH scheme is more accurate, at 99.48%, than the other methods. Compared to the other methods that try to provide a partial solution, our proposed HUDH scheme is a complete solution that prevents unauthorized access of cloud computing.

**Table 3.** Comparisons with existing methods.

| Method | IDS | Encryption | Access Control |
|:---:|:---:|:---:|:---:|
| [46] | ✓ | - | - |
| [35] | - | - | ✓ |
| [42,49] | - | ✓ | ✓ |
| Ours | ✓ | ✓ | ✓ |

We also chose to adopt a light symmetric encryption algorithm (i.e., AES) to avoid the expensive computations required for other public cryptography-based related methods. Asymmetric algorithms are slower than symmetric algorithms, and some solutions suggested using cloud servers for decryption, which is impractical in some situations such as IoT environments.

Moreover, unlike some related methods that use rule-based approaches (i.e., requiring an expert who is responsible for designing the hand-crafted IDS rules), the proposed IDS is automated (i.e., based on machine learning) and achieved decent results, including high accuracy as indicated in the experimental results section.

## 7. Conclusions

In this paper, we introduced the HUDH scheme, which combines three state-of-the-art algorithms (AES, ABAC, and HIDS) for improving data access control in cloud computing. The ABAC and AES algorithms are implemented using JSON files. The proposed HUDH scheme significantly improves data security and user authentication compared to other state-of-the-art methods. The intrusion detection accuracy is improved using the features of the neural network algorithm. One of the advantages of using the proposed algorithm is reducing the possibility of unauthorized data access. We implemented the ABAC and AES algorithms in Java and HIDS on Python. Th HIDS scheme obtains features from two algorithms: first, the random forest algorithm is used for the selection of important features; second, the neural network model is used to train the data.

The HIDS scheme also detects intrusion and the accuracy for Classes 4 and 6 were determined as 0.9710 and 0.9975, respectively. Moreover, the proposed HUDH scheme can enable the data owner to delegate most of the computation overhead to powerful cloud servers. Confidentiality of user access privilege and user secret key accountability are achieved. The formal security proofs showed that the proposed scheme is secure under standard cryptographic models. In the future, we will compare the HUDH scheme with similar types of state-of-the-art mechanisms using different quality of service parameters.

**Author Contributions:** A.R. and N.S., conceptualization, writing, idea proposal, methodology, and results; B.A. and M.A., conceptualization, draft preparation, editing, and visualization; A.M., writing and reviewing; A.A. draft preparation, editing, and reviewing. All authors have read and agreed to this version of the manuscript.

**Funding:** This work was partially supported by the Sensor Networks and Cellular System (SNCS) Research Center under grant 1442-002.

**Acknowledgments:** Taif University Researchers Supporting Project number (TURSP-2020/302), Taif University, Taif, Saudi Arabia. The authors gratefully acknowledge the support of the SNCS Research Center at the University of Tabuk, Saudi Arabia. In addition, the authors would like to thank the dean of scientific research at Shaqra University for supporting this work.

**Conflicts of Interest:** The authors declare no conflict of interest.

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
