# Peer review of "Big Data Handling Approach for Unauthorized Cloud Computing Access"

_electronics, doi:10.3390/electronics11010137_

Round 1
Reviewer 1 Report
The following needs to be considered and alterations should be made to the paper accordingly.
1. How practicable is your proposed method?
2. Is it cost-efficient?
3 . There is insufficient explanation of diagrams and tables used.
4. Are the numbers and formulas generic or generated by you?
5. Grammar requires re-editing
Consider the following papers for proper supplementing and clarity:
• doi.org/10.3390/electronics10172110.
• doi: 10.1109/ACCESS.2021.3049564.
Good organization of paper. Good potential with fairly impressive experimental results.
Reviewer 2 Report
This paper proposes a hybrid unauthorized data handling (HUDH) scheme for Big data in cloud computing. The HUDU aims to restrict illegitimate users from accessing the cloud and data security provision. The proposed HUDH consists of three steps: data encryption, data access, and intrusion detection. HUDH involves three algorithms; Advanced Encryption Standards (AES) for encryption, Attribute-Based Access Control (ABAC) for data access control, and Hybrid Intrusion Detection (HID) for unauthorized access detection. The proposed scheme is implemented using Python and Java language. Testing results demonstrate that the HUDH can delegate computation overhead to powerful cloud servers. User confidentiality, access privilege, and user secret key accountability can be attained with more than 97% high accuracy.
The scheme is presented in a comprehensive form. The authors need to do following revisions:
- Define each notation/symbol used in the algorithms
- Moreover, analyze the complexity of each algorithm
- What does n denote in Eq. 12?
- Some English sentences needs proper revision. For example, in implementation section (section 5 first paragraph), the sentence “In table 1 shown used materials during the implementation.” Is ambiguous.
- In figure 5b, the confusion matrix is blur. Redraw it or present it in the form of a table.
- Improve the resolution of figure 6.
- Which features the authors discuss in figure 6a, 6b. Explain the feature set.
- Compare the HUDH scheme with other mechanisms using different Quality of service parameters.
Round 2
Reviewer 2 Report
The reviewers have addressed all of my concerns well.